# Gender-Specific Renoprotective Pathways in αMUPA Transgenic Mice Subjected to Acute Kidney Injury

**DOI:** 10.3390/ijms25063544

**Published:** 2024-03-21

**Authors:** Heba Abd Alkhaleq, Shadi Hamoud, Israel Hacker, Tony Karram, Ahmad Fokra, Aviva Kabala, Zaid Abassi

**Affiliations:** 1Department of Physiology and Biophysics, Rappaport Faculty of Medicine, Technion-Israel Institute of Technology, P.O. Box 9649, Haifa 31096, Israel; heba.abd.alkhaleq@gmail.com (H.A.A.); yisraelhacker@gmail.com (I.H.); avivak@technion.ac.il (A.K.); 2Department of Internal Medicine E, Rambam Health Care Campus, Haifa 3109601, Israel; s_hamoud@rambam.health.gov.il; 3Department of Vascular Surgery, Rambam Health Care Campus, Haifa 3109601, Israel; t_karram@rambam.health.gov.il; 4Laboratory Medicine, Rambam Health Care Campus, Haifa 3109601, Israel

**Keywords:** αMUPA transgenic mice, Urokinase-type plasminogen activator, acute kidney injury, gender, renoprotection, endothelial nitric oxide synthase, inflammation

## Abstract

Acute kidney injury (AKI) is a serious health concern with high morbidity and high mortality worldwide. Recently, sexual dimorphism has become increasingly recognized as a factor influencing the severity of the disease. This study explores the gender-specific renoprotective pathways in αMUPA transgenic mice subjected to AKI. αMUPA transgenic male and female mice were subjected to ischemia–reperfusion (I/R)-AKI in the presence or absence of orchiectomy, oophorectomy, and L-NAME administration. Blood samples and kidneys were harvested 48 h following AKI for the biomarkers of kidney function, renal injury, inflammatory response and intracellular pathway sensing of or responding to AKI. Our findings show differing responses to AKI, where female αMUPA mice were remarkably protected against AKI as compared with males, as was evident by the lower SCr and BUN, normal renal histologically and attenuated expression of NGAL and KIM-1. Moreover, αMUPA females did not show a significant change in the renal inflammatory and fibrotic markers following AKI as compared with wild-type (WT) mice and αMUPA males. Interestingly, oophorectomized females eliminated the observed resistance to renal injury, highlighting the central protective role of estrogen. Correspondingly, orchiectomy in αMUPA males mitigated their sensitivity to renal damage, thereby emphasizing the devastating effects of testosterone. Additionally, treatment with L-NAME proved to have significant deleterious impacts on the renal protective mediators, thereby underscoring the involvement of eNOS. In conclusion, gender-specific differences in the response to AKI in αMUPA mice include multifaceted and keen interactions between the sex hormones and key biochemical mediators (such as estrogen, testosterone and eNOS). These novel findings shed light on the renoprotective pathways and mechanisms, which may pave the way for development of therapeutic interventions.

## 1. Introduction

Reperfusion of the blood flow and re-oxygenation of an organ following a brief, unexpected reduction in blood and oxygen levels (hypoxia) are referred to as ischemia–reperfusion injury (IRI). This process results in a deficiency of nutrients and oxygen, subsequently leading to adverse effects, including inflammation and oxidative stress [1]. Reperfusion injury is recognized as a significant cause of acute kidney injury (AKI), and it is associated with elevated mortality and morbidity rates among hospitalized patients [2]. AKI can lead to the destruction of tubular epithelial cells, loss of brush border cell polarity, and disruption of cellular adhesion molecules. These effects result in the detachment of epithelial cells and, consequently, kidney dysfunction [3].

Sexual dimorphism is a widely recognized factor contributing to various renal diseases [4,5]. It influences the incidence, prevalence, and progression of kidney disease. Specifically, male animals exhibit a poorer renal prognosis in aging, cystic kidney disease, and following renal ablation models, as opposed to female animals who demonstrate a link to renoprotection [6]. One possible explanation for this sexual dimorphism may be attributed to the hormonal environment rather than genetic variations in renal structure and function. This notion aligns with previous studies that have investigated the influence of gonadectomy or sex hormone treatment on the progression of renal diseases [7,8,9,10,11,12]. In the majority of animal models, estrogen has been observed to decelerate the progression of kidney disease, while testosterone has been found to accelerate it [13,14,15]. Similarly, in humans, based on the current data available, renal disease exhibits a slower rate of progression in women compared to men, regardless of the degree of hypertension, hyperlipidemia, or other progression-related variables [16,17]. In experimental models of ischemic AKI, female sex consistently confers protection against the development of renal damage [4,18,19,20,21,22,23]. Notably, in normal female rats, renal vascular resistance is higher than in males, and glomerular hemodynamics exhibit sex-specific differences [24]. This heightens the notion that renal vascular resistance in female rats plays a beneficial role by averting glomerular damage, thus inhibiting the progression of glomerular capillary pressure [24]. In humans, sexual dimorphism is also evident in terms of glomerular hemodynamic responses. When angiotensin II is administered to healthy young-adult males, it results in a higher filtration fraction, indicating increased glomerular capillary pressure. In contrast, women exhibit a reduction in the glomerular filtration rate (GFR) and effective renal blood flow [25]. The difference in response to angiotensin II may protect women from renal complications by limiting increases in glomerular capillary pressure and reducing glomerular hemodynamic stress. This decreased glomerular capillary pressure and reduced hemodynamic stress can slow the progression of kidney disease. Collectively, sex hormones influence essential cellular functions in AKI, leading to sexual dimorphism in IRI. However, it remains uncertain whether this dimorphism is associated with the renoprotective effects of estrogen or the negative impact of testosterone [26].

Nitric oxide (NO) plays a significant role in the progression of renal disease [6,27]. To enhance NO synthesis, endothelial nitric oxide synthase (eNOS) is phosphorylated and activated through the Akt/phosphatidylinositol 3-kinase pathway, which is itself activated by estrogen. The estrogen-induced increase in NO levels leads to a reduction in endothelin overproduction in the post-ischemic kidney, thereby preventing vasoconstriction. The failure of estrogen to attenuate IRI in eNOS mutant mice provides evidence of the essential role of NO in mediating the protective effects of estrogen in this model [23].

I/R in rats was associated with increased renal NO levels in females when compared with males, along with a smaller increase in peroxynitrite and 3-nitrotyrosine concentration in females than in males. Pretreatment with the antioxidants N-acetyl-L-cysteine or ebselen abolished these sex differences, suggesting that a greater oxidative and nitrosative stress worsens renal damage in males [28]. Estradiol, in particular, displays protective effects against endothelial damage, while testosterone is shown to exacerbate the damage caused by reactive oxygen species (ROS) production [29]. Testosterone leads to an exaggerated inflammatory response in men, potentially rendering them more susceptible to IRI. Notably, male mice subjected to castration demonstrated a reduced production of inflammatory cytokines following ischemia [20]. 

In the current study, alpha murine urokinase-type plasminogen activator (αMUPA) transgenic mice, whose phenotype is similar to that of calorie-restricted (CR) animals, spontaneously consumed 20% less food and lived 20% longer than their wild type (WT). Additionally, αMUPA mice keep a youthful appearance even as they become older [30]. αMUPA mice were originally created as a tool to study the potential involvement of uPA in the eye’s pathophysiology [31,32]. However, analysis of the body organs for urokinase plasminogen activator (uPA) revealed that it is also overexpressed in the brain, which affects their appetite, food intake and leptin levels and subsequently may affect the behavior of other organs, including the kidney [33,34]. In addition, experimental studies showed that αMUPA mice were protected against myocardial infarction (MI) [34,35] and sepsis [36]. Furthermore, we have demonstrated that αMUPA transgenic mice, especially females, exhibited attenuated renal injury/dysfunction in response to AKI, as was evidenced by the lower serum creatinine (SCr) and blood urea nitrogen (BUN), normal renal histologically and attenuated expression of neutrophil gelatinase-associated lipocalin (NGAL) and kidney injury molecule (KIM-1) [37]. Of notice, αMUPA females did not show a significant change in renal inflammatory and fibrotic markers following AKI as compared with wild-type (WT) mice and αMUPA males. The mechanisms underlying the resistance of αMUPA females to AKI are not fully characterized. We hypothesize that sex hormones play a role in this phenomenon, where estrogen exerts a beneficial and testosterone an adverse role in an I/R-induced AKI model. In the current study, αMUPA mice were utilized to elucidate endogenous pathways that promote gender-specific renoprotection in response to AKI. 

## 2. Results

### 2.1. Impact of I/R-Induced AKI following Gonadectomy or L-NAME Treatment on Body and Kidney Weight

αMUPA female mice exhibit a lower basal body weight (BW) than males (Table 1). The BW of αMUPA females remains unaffected by AKI when compared to their male counterparts prior to any treatment. In contrast, αMUPA males experience a significant (−8.34%) reduction in BW following AKI. In oophorectomized females, AKI resulted in a significant (−8.2%) decrease in the BW of αMUPA females, while in orchiectomized males, AKI led to a lesser reduction (−5.9%). L-NAME treatment led to a significant decrease in body weight (BW) in both male (−6.97%) and female (−7.9%) αMUPA mice compared to their initial BW before L-NAME treatment, with a more pronounced reduction observed in αMUPA females (Table 1). Both αMUPA male and female mice exhibited an increase in kidney weight (KW) following AKI compared to their respective sham counterparts (Table 1). Notably, αMUPA female mice showed only a slight change in KW after AKI, in contrast to the significant increase observed in αMUPA males (Table 1). These distinctions persisted even after normalizing KW to BW (Table 1). The KW/BW ratio increased in αMUPA females (from 0.62 ± 0.02% to 0.68 ± 0.02%, *p* < 0.05) following AKI in oophorectomized mice. In contrast, induction of AKI in males that underwent orchiectomy resulted in a significant decrease in the KW/BW ratio in αMUPA males (from 0.79 ± 0.02% to 0.6 ± 0.003%, *p* < 0.001). L-NAME treatment resulted in a significant increase in KW/BW in αMUPA males (0.76 ± 0.02, compared to 0.68 ± 0.02, *p* < 0.05) and females (0.68 ± 0.02, compared to 0.62 ± 0.02, *p* < 0.05) compared with the KW/BW ratio following AKI in the absence of any treatment. 

### 2.2. Kidney Function and Renal Injury Biomarkers

Induction of AKI significantly increased the SCr and BUN levels 48 h post-injury in αMUPA untreated males (Figure 1A,B). In contrast, the SCr and BUN levels in αMUPA untreated females were not significantly changed in response to I/R-AKI. The SCr and BUN levels in αMUPA females remained unaffected even following oophorectomy. Notably, orchiectomy performed 2 weeks prior to AKI resulted in a reduction in the SCr and BUN levels in αMUPA males (Figure 1A,B). Similar results were observed after three days of L-NAME treatment (Figure 1E,F). αMUPA untreated male mice exhibited a significant increase in renal expression of both NGAL and KIM-1, biological markers of AKI, following AKI induction (Figure 1C,D). In contrast, renal expression of NGAL and KIM-1 in αMUPA untreated females remained unchanged after AKI. Notably, the renal expression of NGAL and KIM-1 in αMUPA females, which remained unaffected following AKI, was significantly enhanced in the presence of oophorectomy (*p* < 0.01) (Figure 1C,D) and following L-NAME treatment (Figure 1G,H). Conversely, orchiectomy resulted in a significant decrease in renal expression of NGAL and KIM-1. L-NAME treatment also resulted in a reduction of renal NGAL and KIM-1 expression, albeit to a lesser extent when compared to the effect observed with orchiectomy. These results underscore that αMUPA female mice demonstrate increased tolerance to ischemic renal stress facilitated by estrogen-mediated eNOS, as evidenced by their resistance to I/R-AKI, which was abolished by oophorectomy and L-NAME treatment. On the other hand, testosterone heightens sensitivity to AKI–renal injury in αMUPA males, a response that dissipates following orchiectomy.

### 2.3. Renal Histology

Figure 2 illustrates renal histological changes in αMUPA male and female mice subjected to I/R-AKI, both before and after gonadectomy or L-NAME treatment. αMUPA male kidney tissue displayed more pronounced injury as evident by tubular collapse, loss of the brush border, and cellular detachment from tubular basement membranes, prior to any treatment (second column of Figure 2A). Necrosis and the presence of hyaline casts were observed in the outer medulla, with more intense congestion noted in the inner medulla. In contrast, αMUPA female mice demonstrated attenuated kidney injury in response to I/R-AKI, resembling the appearance of sham kidney tissues (second column of Figure 2B). The observed resistance of αMUPA females to I/R-AKI was nullified upon oophorectomy or L-NAME treatment, where these animals displayed exacerbated renal injury, characterized by tubular collapse, loss of the brush border, and cellular detachment from tubular basement membranes (fourth and sixth columns of Figure 2B). Following orchiectomy, αMUPA males exhibited reduced renal damage (fourth column of Figure 2A).

### 2.4. Renal uPA and uPA Receptor (uPAR)

Untreated αMUPA male mice displayed a reduction in renal uPA immunoreactivity (Figure 3D) but not in uPA expression (Figure 3A). Simultaneously, there was a significant elevation in uPA receptor expression (approximately 12-fold) following AKI (Figure 3B). Conversely, renal uPA and uPA receptor expression in αMUPA females remained unaltered following AKI (Figure 3A,B). Notably, female mice exhibited lower basal uPAR immunoreactivity compared to male mice (Figure 3E). Analysis of renal plasminogen activator inhibitor 1 (PAI-1), a specific glycoprotein that inhibits the proteolytic action of uPAR-bound uPA, revealed an increase in αMUPA male mice following AKI (Figure 3C). In contrast, αMUPA female mice did not exhibit significant changes in renal PAI-1 expression following AKI but before treatment. Orchiectomy did not affect renal uPA expression in αMUPA males after AKI (Figure 3A), yet there was a significant increase in uPA immunoreactivity post-orchiectomy (Figure 3D). Furthermore, renal uPAR and PAI-1 expression decreased in αMUPA males after orchiectomy (Figure 3B,C). Conversely, αMUPA female mice underwent oophorectomy before AKI induction displayed a significant elevation in renal uPA expression/abundance (Figure 3A,D), accompanied by a substantial increase in uPA receptor abundance (~9 folds) (Figure 3E). Similar results were observed following L-NAME treatment (Figure 3F,I,J).

### 2.5. Renal Leptin, Insulin Receptor and Peroxisome Proliferator-Activated Receptor-gamma Coactivator (PGC1α)

αMUPA mice (both males and females) exhibited a significant increase in renal leptin expression, a factor previously identified for its protective role against AKI, following the induction of AKI compared to their corresponding sham mice prior to treatment (Figure 4A). In contrast, αMUPA male mice displayed reduced renal insulin receptor (InsR) expression, a crucial element for glomerular and tubular function, compared to sham male animals. In contrast, InsR expression in αMUPA female mice remained unchanged following AKI and maintained levels comparable to those of healthy sham controls. The expression of PGC-1α, a key player in cellular recovery following I/R injury, decreased in αMUPA male mice following AKI, prior to the initiation of any treatment. In contrast, αMUPA female mice did not exhibit a change in renal PGC-1α expression after AKI. αMUPA female mice displayed the highest expression levels of PGC-1α among the four studied subgroups, both at baseline and following AKI. However, αMUPA females experienced a significant reduction in renal leptin (Figure 4A) and PGC1α (Figure 4C) following oophorectomy compared to αMUPA mice without oophorectomy. L-NAME treatment resulted in a massive reduction in renal leptin, InsR and PGC1α expression in both male and female αMUPA mice (Figure 4D–F).

### 2.6. Renal Expression of Inflammatory and Fibrotic Markers

αMUPA male mice showed a significant increase in renal Interleukin 6 (IL-6) and Toll-like receptor 4 (TLR4) expression, proinflammatory cytokines, 48 h following AKI and prior to any treatment (Figure 5A,B). In contrast, αMUPA females did not display a significant change in renal IL-6 following AKI, while TLR4 expression decreased in αMUPA females following renal injury (Figure 5A,B). Analysis of the renal immunoreactive levels of signal transducer and activator of transcription *3* (STAT3), p-STAT3, I Kappa B (IKB) and mitogen-activated protein kinase (MAPK), all pro-inflammatory markers, of untreated αMUPA female mice showed they were not affected by AKI (Figure 5C–F). Moreover, AKI induction provoked renal transforming growth factor β (TGFβ) expression in male αMUPA mice, but not in female mice (Figure 5G). αMUPA mice (both males and females) that were subjected to AKI did not exhibit a significant change in renal IL-6 and TLR4 following gonadectomy (Figure 5A,B). However, they showed a decrease in renal IL-6 and TLR4 expression following L-NAME treatment (Figure 5H,I). Renal immunoreactive expression of p-STAT3, a pro-inflammatory marker, were elevated in αMUPA male mice following orchiectomy (Figure 5D). Notably, αMUPA female mice, which did not show elevations of STAT3 and p-STAT3 following AKI in the absence of oophorectomy, displayed significant increases in these markers when oophorectomized (Figure 5C,D). Similar results were obtained following L-NAME treatment (Figure 5J,K). These results indicate that αMUPA mice, particularly females, lose their resistance to AKI at the inflammatory level in the absence of estrogen by oophorectomy and in the absence of eNOS by L-NAME. Gonadectomy prior to AKI increased the renal IKB and MAPK abundance in αMUPA male and female mice (Figure 5E,F). Moreover, L-NAME treatment increased these reactive protein levels in αMUPA mice (Figure 5L,M). The analysis of renal TGFβ expression in αMUPA mice revealed an exaggerated increase in oophorectomized and L-NAME treated females, which was more remarkable than in orchiectomized and L-NAME treated males (Figure 5G,N). These results highlight the beneficial role of estrogen and eNOS in nephroprotection against AKI-induced fibrosis in αMUPA female mice. Moreover, testosterone appears to increase renal sensitivity to AKI at the fibrotic level.

### 2.7. Renal Apoptotic and Autophagy Markers

In response to AKI, and in the absence of any treatment, αMUPA male mice exhibited an increase in renal expression of the apoptotic marker Caspase 7 compared to their sham counterparts (Figure 6B). However, kidney expression of Caspase 3 and Caspase 7 did not change in αMUPA female mice following AKI (Figure 6A,B). Moreover, the renal expression of various autophagy markers, including microtubule-associated protein 1A/1B-light chain 3 (LC3), P62, and Galectin 8, were notably low in αMUPA female mice under both normal conditions and following AKI (Figure 6C–F). Interestingly, αMUPA male mice showed a significant increase in Caspase 3 in response to AKI induction in orchiectomized compared to untreated αMUPA male mice (Figure 6A). Caspase 7 expression increased significantly following oophorectomy in female αMUPA mice (Figure 6B). Both αMUPA male and female mice displayed an elevation in renal expression of autophagy markers, including LC3, P62 and Galectin 8 following orchiectomy (in males) and oophorectomy (in females) compared with their AKI-operated untreated αMUPA counterparts (Figure 6C–F). L-NAME treatment exaggerated the expression of renal apoptotic and autophagic markers in both male and female αMUPA mice (Figure 6G–L).

### 2.8. Renal Angiotensin-Converting Enzyme 2 (ACE2) and Mas Receptor (MasR)

In untreated mice, αMUPA males and females did not exhibit a significant decrease in the expression of renal ACE2 and MasR following AKI (Figure 7A,B). This suggests that ACE2 may play a role in the protective pathway of αMUPA kidneys following AKI. In addition, the expression of renin, the enzyme that plays an essential role in the RAAS, was lower in αMUPA female mice and remained unaffected by AKI (Figure 7C). Conversely, αMUPA male mice exhibited a significant reduction in renal ACE2 and MasR expression, as well as ACE2 immunoreactivity, when orchiectomy took place prior to AKI (Figure 7A,B,D). The expression of renin enhanced following orchiectomy + AKI as compared with AKI-operated male mice without orchiectomy (Figure 7C). In αMUPA female mice, the expression of renal MasR decreased and the expression of renin increased significantly in response to AKI in the presence of oophorectomy (Figure 7B,C). L-NAME treatment resulted in a statistical decrease in renal ACE2 and MasR expression and ACE2 immunoreactivity in male and female mice, but more remarkably in females (Figure 7E,F,H). L-NAME treatment prior to AKI increased the renal expression of renin in both sexes, especially the females (Figure 7G).

### 2.9. Renal Endothelial Nitric Oxide Synthase (*eNOS*)

Untreated αMUPA female mice exhibited higher levels of renal eNOS mRNA (Figure 8C). Both αMUPA males and females showed an increase in eNOS immunoreactivity following AKI compared with sham-operated mice (Figure 8A). However, αMUPA mice did not show any change in the p-eNOS immunoreactive protein levels before (sham) and after AKI (Figure 8B). AKI-operated males that underwent orchiectomy showed a decrease in eNOS immunoreactivity concomitant to an increase in the p-eNOS protein levels under both sham and AKI conditions (Figure 8A,B). Additionally, these mice exhibited enhanced renal eNOS expression (Figure 8C). Oophorectomized females showed an augmented increase in the eNOS and p-eNOS immunoreactive protein levels, along with a significant decrease in renal eNOS expression following AKI (Figure 8A–C). Collectively, the estrogen-related preservation of αMUPA female mouse kidneys following AKI was influenced by the eNOS pathway. L-NAME treatment statistically increased the abundance of eNOS and resulted in absence of activated eNOS (p-eNOS) in male and female αMUPA mice (Figure 8D–F).

## 3. Discussion

Renal ischemia, or reduced blood flow to the kidneys, is the primary cause of acute kidney injury (AKI). Furthermore, renal hypoxia, a related pathological condition, is caused by reduced oxygen supply to the kidneys [38]. It ultimately decreases the glomerular filtration rate (GFR) and renal output, and it increases the morbidity and mortality rates [38]. A recognized feature of chronic progressive kidney disease is sexual dimorphism, with female sex being associated with renoprotection [29]. Although less commonly known, it is worth noting that sexual dimorphism has also been linked to the development of nephrotoxic and ischemic acute kidney damage (AKI) [5]. Experimental models of ischemic AKI in animals have shown that female sex provides protection against the development of renal damage [5,18,19,20,21,22,28,39,40,41]. Viñas et al. demonstrated that female mice were resistant to ischemic renal injury compared with males, as determined by the serum Cr and NGAL, histologic scores, neutrophil infiltration, and extent of apoptosis [42]. Similar findings were shown in our previous study, where AKI induction resulted in reduced renal damage in αMUPA animals, particularly in females, as seen by the decreased serum creatinine (SCr), serum blood urea nitrogen (BUN), and kidney NGAL and KIM-1 expression [37]. The mechanisms behind αMUPA females’ resilience to AKI are unknown. We hypothesize that sex hormones are involved in these phenomena, with estrogen having a positive effect and testosterone having a negative effect in the I/R-induced AKI model. To test this hypothesis, male mice underwent orchiectomy (testes removal process to block the testosterone synthesis) and female mice underwent oophorectomy (ovaries removal process to block the estrogen synthesis). In line with this hypothesis, the current study demonstrated that oophorectomy in females eliminated their resistance to I/R renal injury, highlighting the central protective role of estrogen. Correspondingly, orchiectomy in males mitigated their exaggerated renal damage in response to AKI, thereby emphasizing the deleterious effects of testosterone. Additionally, treatment with L-NAME abolished αMUPA females’ resistance to AKI, underscoring the involvement of eNOS in this phenomenon.

Previous studies have demonstrated that calorie restriction (CR), involving a 20–40% reduction in food intake, could ameliorate both acute kidney injury (AKI) and chronic kidney disease (CKD) [43,44]. αMUPA transgenic mice are calorically restricted mice, consuming smaller amounts of food compared to their wild-type (WT) FVB/N counterparts [33]. Thus, these αMUPA mice offer numerous advantages, including increased lifespan, reduced body weight and fat mass, lower body temperature, and improved insulin sensitivity. Additionally, they exhibit decreased blood levels of insulin-like growth factor-1 (IGF-1) and a reduced incidence of naturally occurring malignancies and tumorigenic lesions [33,45,46,47]. The paramount advantage of these αMUPA mice, with a particular emphasis on females, became evident in studies highlighting cardioprotection following myocardial infarction (MI). Female αMUPA mice exhibited a notable reduction in cardiac aging, fractional shortening, infarct size, and the post-MI inflammatory response [34]. Additionally, other studies showed lower expression of pro-inflammatory genes in the hearts of αMUPA females compared to their wild-type (WT) counterparts after MI and LPS (lipopolysaccharide) treatment [35,36]. Previously, we have reported that αMUPA transgenic mice, especially females, exhibited attenuated renal injury/dysfunction in response to AKI, as was evidenced by the lower serum creatinine (SCr) and blood urea nitrogen (BUN), normal renal histologically and attenuated expression of NGAL and KIM-1 [37]. Of notice, αMUPA females did not show a significant change in renal inflammatory and fibrotic markers following AKI as compared with wild-type (WT) mice and αMUPA males. The present study provides new insights into the mechanisms underlying the resistance of αMUPA females to AKI, where we demonstrated that estrogen plays a beneficial and testosterone an adverse role in an I/R-induced AKI model. Furthermore, our findings implicate NO as a key mediator of gender-specific renoprotection.

Our results indicate that oophorectomy, which results in estrogen deprivation, reversed the post-ischemic renal protection observed in αMUPA females. This reversal was evidenced by the decreased body weight, increased kidney/body weight ratio and augmented the renal damage indicated by renal expression of NGAL and KIM-1, markers of renal injury. Furthermore, at the histological level, we can observe injuries characterized by tubular collapse, loss of the brush border, and cellular detachment from tubular basement membranes in oophorectomized females subjected to AKI. In contrast, orchiectomy reduced the testosterone blood levels, thereby attenuating post-ischemic renal damage. This was characterized by a decreased body weight change ratio (compared with untreated AKI-operated males), as well as the reduction of all renal injury markers, including SCr, BUN, and the renal expression of NGAL and KIM-1. 

Functional renal failure in AKI is caused by damaged renal epithelial tubular cells as a result of ischemic kidney injury. Although AKI is characterized by tubular cell death, there is increasing evidence that renal ischemia also affects endothelial cell function [19]. The latter plays an important role in renal tubular epithelial damage and eventually the pathophysiology of ischemic kidney disease [48]. Specifically, endothelial damage is characterized by downregulation of eNOS in association with impaired availability of NO, thus worsening the inflammatory response [49], thereby promoting the disease. Estrogen may moderate the gender variations in renal I/R by influencing the eNOS expression and production of NO [4]. In agreement with this concept, we demonstrated that oophorectomy abolished their remarkable resistance to AKI-induced renal damage. Moreover, the importance of eNOS in this matter was addressed via blocking the eNOS activity by L-NAME in αMUPA male and female mice. L-NAME treatment aggravated the changes in body weight and kidney weight following AKI, highlighting the crucial role of eNOS as a protective mediator against AKI, especially in females, which show exaggerated results similar to the results following oophorectomy. Notably, both orchiectomy and oophorectomy elevate the renal immunoreactive levels of uPA and uPAR in αMUPA mice, with an exaggerated level in females. Similar results were observed following L-NAME treatment. This indicates that the estrogen-related eNOS axis plays a nephroprotective role in females, as well as eNOS (don’t think eNOS is needed) in males, where elimination/inhibition of this enzyme exerts adverse effects in αMUPA mice following gonadectomy or L-NAME treatment.

Leptin, InsR and PGC1α, the renal protective mediators, decreased following oophorectomy but were not affected by orchiectomy. These results indicate that estrogen, not testosterone, affects the expression of protective markers in female kidneys. L-NAME treatment negatively affected the renal expression of leptin, InsR, and PGC1α in both males and females, suggesting that eNOS is one of the mediators in αMUPA protective kidneys. 

At the inflammatory level, oophorectomy leads to an increase in all the proinflammatory markers, including STAT-3, p-STAT3, IKB, and MAPK immunoreactivity. Conversely, orchiectomy increases these markers to a lesser extent. L-NAME treatment yielded similar results, indicating the involvement of estrogen-mediated eNOS in the protective response of αMUPA females’ kidneys against the inflammatory process. L-NAME treatment also affects the male kidneys to a greater extent than orchiectomy.

Angiotensin II and transforming growth factor β (TGF-β) undergo upregulation in renal diseases and are well known to play a causative role in its progression. TGF-β expression in the glomeruli, podocytes and throughout the kidneys is reduced by estrogen and increased by testosterone in experimental models of kidney disease [29], a finding that may explain the currently observed results. Noteworthy, estrogen reduces fibrosis generation following renal injury by affecting multiple factors. For instance, estradiol inhibits the production of type I collagen in mesangial cells [50] and suppresses type IV collagen synthesis mediated by transforming growth factor (TGF) β1 [51], thus attenuating the fibrotic process characterizing renal diseases. Interestingly, both oophorectomy and L-NAME treatment increases TGF-β expression in females but not in males. This suggests a protective role of estrogen-related eNOS against AKI in αMUPA females at the fibrotic level. 

Autophagy and apoptotic marker expression was elevated following gonadectomy and L-NAME treatment. These findings prove that eNOS, estrogen, and testosterone affect renal injury at the autophagic and apoptotic levels. ACE2 immunoreactivity in males significantly decreased following AKI, along with eNOS inhibition, more than the effect of orchiectomy, where the ACE2 abundance did not change. However, in females, L-NAME treatment and oophorectomy showed similar results. MasR decreased and renin increased following gonadectomy and L-NAME treatment. The pronounced elevation of eNOS in female kidneys following AKI may explain the greater degree of injury following AKI in the absence of estrogen or eNOS. After gonadectomy, there is a decrease in eNOS abundance and an increase in p-eNOS in both sexes to protect their kidneys from renal injury after AKI, with a more pronounced change in females. After L-NAME treatment, there is a dramatic increase in the immunoreactivity of eNOS, while the p-eNOS protein level was completely eliminated. This supports the notion that L-NAME treatment effectively blocks the reactivity of eNOS in αMUPA mice. 

The current study has a few limitations. 1—The applied model, I/R-AKI, represents only one subtype of AKI, namely ischemic kidney injury; therefore, our conclusions are for this model but not for nephrotoxic or septic-AKI. 2—The applied ischemic duration is 30 min, so it is appealing to compare it to 45 min of renal artery occlusion. 3—Gonadectomy does not always achieve complete elimination of sex hormones. 4—The sample size is moderate yet sufficient to achieve statistical significance, which meets the ethical requirements. 

Collectively, αMUPA female mice lost their renal resistance mechanisms against I/R AKI following oophorectomy, and the reduction in estrogen levels in these animals exacerbated their susceptibility to renal ischemic injury, defining the characteristics of AKI. Additionally, L-NAME treatment in both male and female mice negatively affected the crucial protective role of renal eNOS during AKI. These novel findings shed light on the renoprotective pathways and mechanisms, which may pave the way for development of therapeutic interventions.

## 4. Materials and Methods

### 4.1. Animals 

αMUPA transgenic mice were sourced from Prof. Ruth Miskin at the Weizmann Institute of Science (Rehovot, Israel) and were subsequently bred at the Rappaport Faculty of Medicine (Haifa, Israel). The mice were housed in groups of five per cage in a facility with controlled temperature conditions and were provided with standard mouse chow. All the experiments were conducted in strict adherence to the guidelines established by the Committee for the Supervision of Animal Experiments at the Technion, Israel Institute of Technology (IL 097-06-19, IL-191-12-23IL). The sample size was determined based on statistical power, namely to achieve 80% power with α = 0.05.

Gonadectomy surgeries: αMUPA males (*n* = 14) underwent bilateral orchiectomy, where the surgical removal of both testes was performed two weeks before the induction of AKI or sham operation. The males were anesthetized using a mixture of 100 mg/kg ketamine and 10 mg/kg xylazine, and they were placed in a dorsal position on a controlled thermoregulated table to maintain a core temperature of 37 °C. A midline incision was made on the ventral side of the scrotum, the tunica was pierced, and each testis was gently exposed. After disconnecting the vas deferens and blood vessels by ligation, the testes were extracted. The abdomen was then sutured, and the animals were returned to their cages for a 2-week recovery period prior to sham or AKI induction. αMUPA females (*n* = 12) underwent bilateral oophorectomy, which involved surgical removal of both ovaries, two weeks before the induction of AKI or sham operation. The females were anesthetized as described above. An incision was made in the abdominal wall, the fallopian tubes were exposed, and each ovary was isolated from the blood supply and surrounding tissues by ligation. The two ovaries were removed, the abdomen was sutured, and the animals were returned to their cages for a 2-week recovery period before sham or AKI induction.

L-NAME treatment: αMUPA males (*n* = 12) and females (*n* = 12) at 12 weeks of age, received N omega-nitro-L-arginine methyl ester hydrochloride (L-NAME) (75 mg/L) added to drinking water three days before sham or AKI induction. 

Induction of AKI: αMUPA male (*n* = 12, weighing 22–35 g) and female (*n* = 12, weighing 17–26 g) mice, that underwent gonadectomy two weeks prior to the experiment or received L-NAME treatment three days prior to sham or AKI were studied. The mice were anesthetized with a mixture of 100 mg/kg ketamine and 10 mg/kg xylazine and placed dorsally on a thermoregulated table to maintain a core temperature of 37 °C. After opening the abdominal wall, the intestines were covered with saline-soaked gauze to minimize dehydration. The renal arteries of the experimental groups were bilaterally exposed and clamped for 30 min to induce ischemia, followed by reperfusion. Post-reperfusion, blood flow was visually verified, the abdomen sutured, and the animals returned to their cages until sacrifice. The reperfusion time, determined from clamp removal, and its impact on the SCr and BUN levels, histological sections, and the expression of relevant biomarkers were examined after 48 h. The sham-operated animals underwent the same procedure without renal artery clamping, serving as controls.

### 4.2. Evaluation of Blood Variables and Renal Functional Parameters

48 h following the AKI procedure, the mice were anesthetized with a mixture of 100 mg/kg ketamine and 10 mg/kg xylazine. Blood samples (500 μL each) were drawn through the left ventricle of the heart and centrifuged at 1500 rpm for 10 min at 4 °C. The serum creatinine (SCr) and BUN levels were determined in the mouse serum using commercial kits (Siemens, Germany) with an auto-analyzer dedicated instrument (Dimension RXL, Siemens, Germany).

### 4.3. Histopathology

Kidneys from the various experimental groups were extracted, fixed in 4% formalin, and subsequently embedded in paraffin. Transverse sections were stained with hematoxylin and eosin (H&E). In brief, the paraffin tissue sections underwent rehydration through standard protocols, involving three rounds of clearing in xylene, followed by rehydration in anhydrous alcohol (100% and 95%) and distilled water. The sections were stained with hematoxylin for 2 min, rinsed with tap water, and blotted dry. Subsequently, the slides were incubated with eosin stain for 2 min, followed by routine dehydration in 95% alcohol and xylene. Finally, the slides were sealed with a slide mounting medium (DPX). The histological analysis of the renal tissue, including assessments for casts, necrosis, and inflammation, was conducted on both WT mice and αMUPA animals subjected to either sham operation or AKI.

### 4.4. Real-Time PCR

Complete RNA was purified from frozen kidney samples (cortex and medulla) using TRIzol reagent. cDNA was synthesized according to the manufacturer’s protocol from complete RNA using the maximal first strand cDNA synthesis kit for RT-qPCR. Using PerfeCTa SYBR Green with the target gene primers, quantitative real-time PCR analysis was performed and analyzed in the 7500 Real Time PCR System (Applied Biosystems, RHENIUM 8440, Foster City, CA, USA). The mRNA levels of the various genes (uPA, uPAR, eNOS, ACE2, IL-6, leptin, LC3, KIM-1, NGAL and peroxisome proliferator-activated receptor-gamma coactivator (PGC1α)) were standardized to mRNA levels of Rpl13a, referred to as the housekeeping gene. Relative to the normalized values obtained for WT mice at baseline, fold shift was measured.

Mouse primers: uPA: -F-AGAGTCTGAAAGTGACTATCTC,-R-CCTTCGATGTTACAGATAAGCuPAR: -F-TCTGGATCTTCAGAGCTTTC,-R-GCCTCTTACGGTATAACTCC PAI-1: -F-AGCAACAAGTTCAACTACAC,-R-CTTCCATTGTCTGATGAGTTC InsR: -F-AAGACCTTGGTTACCTTCTC,-R-GGATTAGTGGCATCTGTTTG eNOS: -F-AAAGCTGCAGGTATTTGATG,-R-AGATTGCCTCTATTTGTTGC ACE2: -F-CATTTGCTTGGTGATATGTG,-R-GCCTCTTGAAATATCCTTTCTG MasR: -F-GTTTAAGGAACTCTGGAAGATG,-R-TTAGTCAGTTAGTCAGTGGC Renin: -F-AGCCAAGGAGAAGAGAATAG,-R-CTCCTGTTGGGATACTGTAG IL-6: -F-GTCTATACCACTTCACAAGTC,-R-TGCATCATCGTTGTTCATAC TLR4: -F-TCCCTGCATAGAGGTAGTTCC,-R-TCCAGCCACTGAAGTTCTGA Leptin: -F-ACATTTCACACACGCAGTCGG,-R-GGACCTGTTGATAGACTGCCA LC3: -F-GAACCGCAGACGCATCTCT,-R-TGATCACCGGGATCTTACTGG P62: -F-AATGTGATCTGTGATGGTTG,-R-GAGAGAAGCTATCAGAGAGG Galectin 8: -F-ATATACAAAAGCCAGGCAAG,-R-CAAATGCTTTCACATTGAGG TGF-β: -F-GGATACCAACTATTGCTTCAG,-R-TGTCCAGGCTCCAAATATAG Caspase 3: -F-CATAAGAGCACTGGAATGTC,-R-GCTCCTTTTGCTATGATCTTC Caspase 7: -F-CAAAACCCTGTTAGAGAAACC,-R-CCATGAGTAATAACCTGGAAC KIM-1: -F-CTGGAGTAATCACACTGAAG,-R-AAGTATGTACCTGGTGATAGC NGAL: -F-ATATGCACAGGTATCCTCAG,-R-GAAACGTTCCTTCAGTTCAG PGC1α: -F-TCCTCTTCAAGATCCTGTTAC,-R-CACATACAAGGGAGAATTGC Rpl13a: -F-AAGCAGGTACTTCTGGGCCG,-R-GGGGTTGGTATTCATCCGCT

### 4.5. Western Blot

Samples of kidney tissue (20 mg), encompassing both cortex and medulla, were homogenized in a lysis buffer, and protein quantification was conducted using the Bradford commercial assay. Protein samples (50 μg) underwent electrophoresis on a denaturing 10% sodium dodecyl sulfate (SDS) polyacrylamide gel, followed by electro-transfer to nitrocellulose membranes for 1.5 h at 100 V. Membranes were blocked with 5% BSA in Tris-buffered saline (TBS) for 1 h at room temperature. Primary antibodies were diluted in TBST at concentrations of 1:200–1:1000 with 5% BSA and incubated overnight at 4 °C. To serve as an internal control, immuno-detection of GAPDH with monoclonal anti-GAPDH antibodies was performed. HRP-secondary antibodies were applied for 45 min at room temperature at a concentration of 1:15,000. The signal was detected with ECL (chemiluminescence substrate), and images were captured using ImageQuant LAS 4000.

### 4.6. Statistical Analysis

Animals were randomly assigned to the experimental group. Results are presented as mean ± SEM. Statistical significance was tested for comparisons between male and female αMUPA mice using unpaired Student’s *t*-tests. A *p* < 0.05 value was found to be statistically significant.

## Figures and Tables

**Figure 1 ijms-25-03544-f001:**
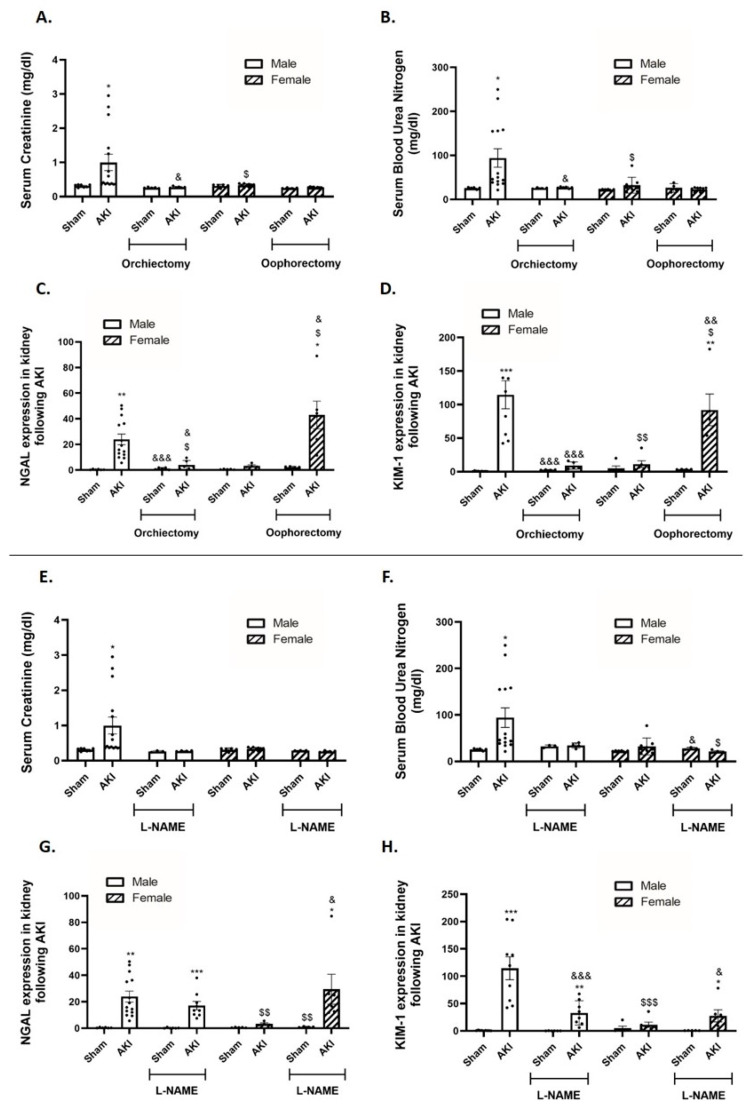
The impact of AKI on kidney function/renal injury in αMUPA male and female mice before and after Gonadectomy/L-NAME treatment. (**A**) Serum creatinine level; (**B**) Blood urea nitrogen (BUN) level; (**C**) Renal expression of NGAL and (**D**) Renal expression of KIM-1 following Orchiectomy (males) or Oophorectomy (females); (**E**) Serum creatinine level; (**F**) Blood urea nitrogen (BUN) level; (**G**) Renal expression of NGAL and (**H**) Renal expression of KIM-1; following L-NAME treatment. * *p* < 0.05, ** *p* < 0.01, *** *p* < 0.001—Sham vs. AKI in the same group, ^$^ *p* < 0.05, ^$$^ *p* < 0.01, ^$$$^ *p* < 0.001—male vs. female, which underwent similar procedure/treatment, ^&^ *p* < 0.05, ^&&^ *p* < 0.01, ^&&&^ *p* < 0.001—before vs. after Orchiectomy (males) or Oophorectomy (females)/L-NAME treatment.

**Figure 2 ijms-25-03544-f002:**
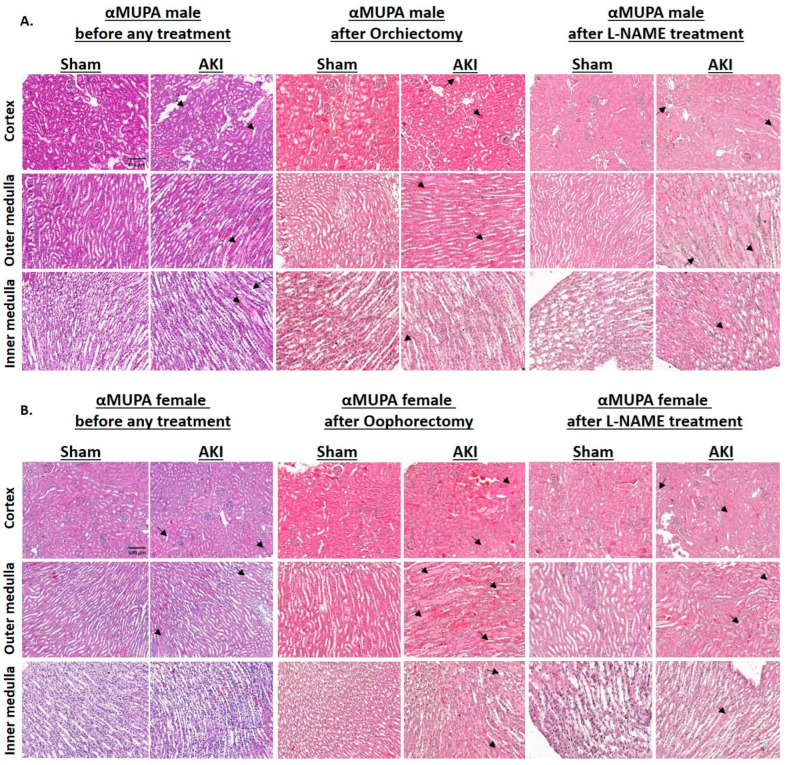
Effects of AKI on renal histology of αMUPA male and female mice before and after gonadectomy/L-NAME treatment. (**A**) Representative cortical, outer medullary and inner medullary histological sections of kidneys stained with hematoxylin and eosin taken from αMUPA male mice: 1—untreated male sham αMUPA mice (first column), 2—untreated male AKI αMUPA mice (second column); 3—male sham αMUPA mice that underwent orchiectomy (third column), 4—male AKI αMUPA mice that underwent orchiectomy (fourth column); 5—male sham αMUPA mice after L-NAME treatment (fifth column), 6—male AKI αMUPA mice after L-NAME treatment (sixth column). (**B**) Renal sections taken from kidneys of αMUPA female mice. The long arrows indicate tubular collapse, loss of the brush border, cellular detachment from tubular basement membranes, the short arrows indicate congestion. Images were taken in X20 magnification, scale bar = 100 µm.

**Figure 3 ijms-25-03544-f003:**
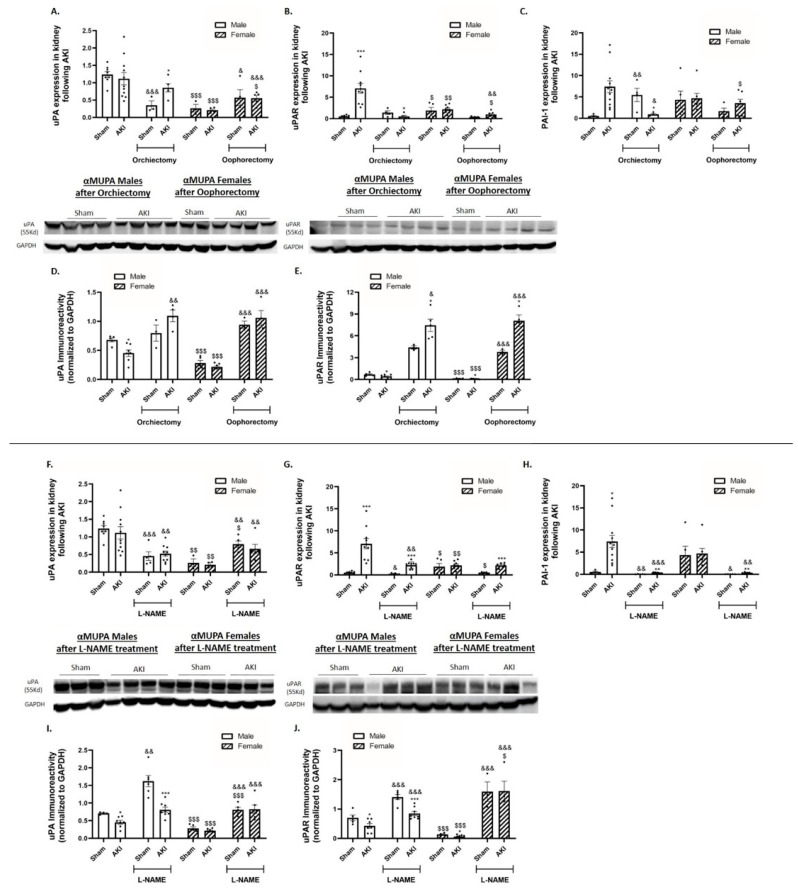
The impact of AKI on renal expression and abundance of uPA and uPAR in αMUPA male and female mice before and after gonadectomy/L-NAME treatment. (**A**) Urokinase plasminogen activator (uPA) expression; (**B**) urokinase plasminogen receptor (uPAR) expression; (**C**) plasminogen activator inhibitor 1 (PAI-1) expression; (**D**) immunoreactive levels of urokinase plasminogen activator (uPA); and (**E**) immunoreactive levels of urokinase plasminogen activator receptor (uPAR) following orchiectomy (males) or oophorectomy (females). (**F**) Urokinase plasminogen activator (uPA) expression; (**G**) urokinase plasminogen receptor (uPAR) expression; (**H**) plasminogen activator inhibitor 1 (PAI-1) expression; (**I**) immunoreactive levels of urokinase plasminogen activator (uPA); and (**J**) immunoreactive levels of urokinase plasminogen activator receptor (uPAR) after L-NAME treatment. (* *p* < 0.05, ** *p* < 0.01, *** *p* < 0.001—sham vs. AKI in the same group, ^$^ *p* < 0.05, ^$$^
*p* < 0.01, ^$$$^ *p* < 0.001—male vs. female, which underwent similar procedure/treatment, ^&^ *p* < 0.05, ^&&^ *p* < 0.01, ^&&&^ *p* < 0.001—before vs. after orchiectomy (males) or oophorectomy (females)/L-NAME treatment).

**Figure 4 ijms-25-03544-f004:**
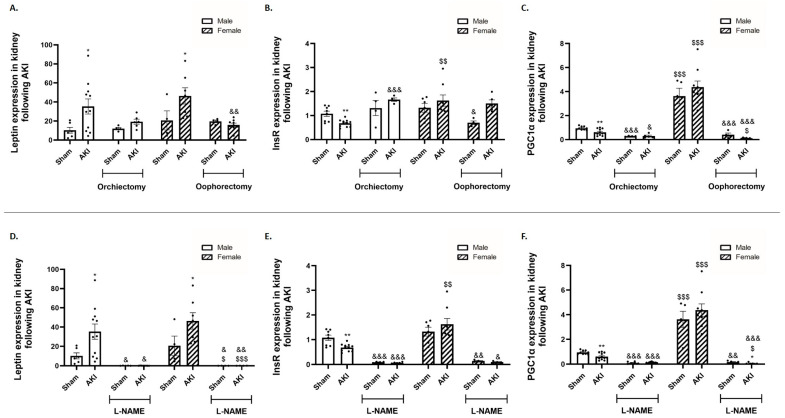
The impact of AKI on the expression of renal leptin, insulin receptor and PGC1-α in αMUPA male and female mice before and after gonadectomy/L-NAME treatment. (**A**) Leptin; (**B**) insulin receptor (InsR); and (**C**) PGC1α following orchiectomy (males) or oophorectomy (females). (**D**) Leptin; (**E**) insulin receptor (InsR); and (**F**) PGC1α following L-NAME treatment. (* *p* < 0.05, ** *p* < 0.01—sham vs. AKI in the same group, ^$^
*p* < 0.05, ^$$^ *p* < 0.01, ^$$$^ *p* < 0.001—male vs. female, which underwent similar procedure/treatment, ^&^ *p* < 0.05, ^&&^ *p* < 0.01, ^&&&^ *p* < 0.001—before vs. after orchiectomy (males) or oophorectomy (females)/L-NAME treatment).

**Figure 5 ijms-25-03544-f005:**
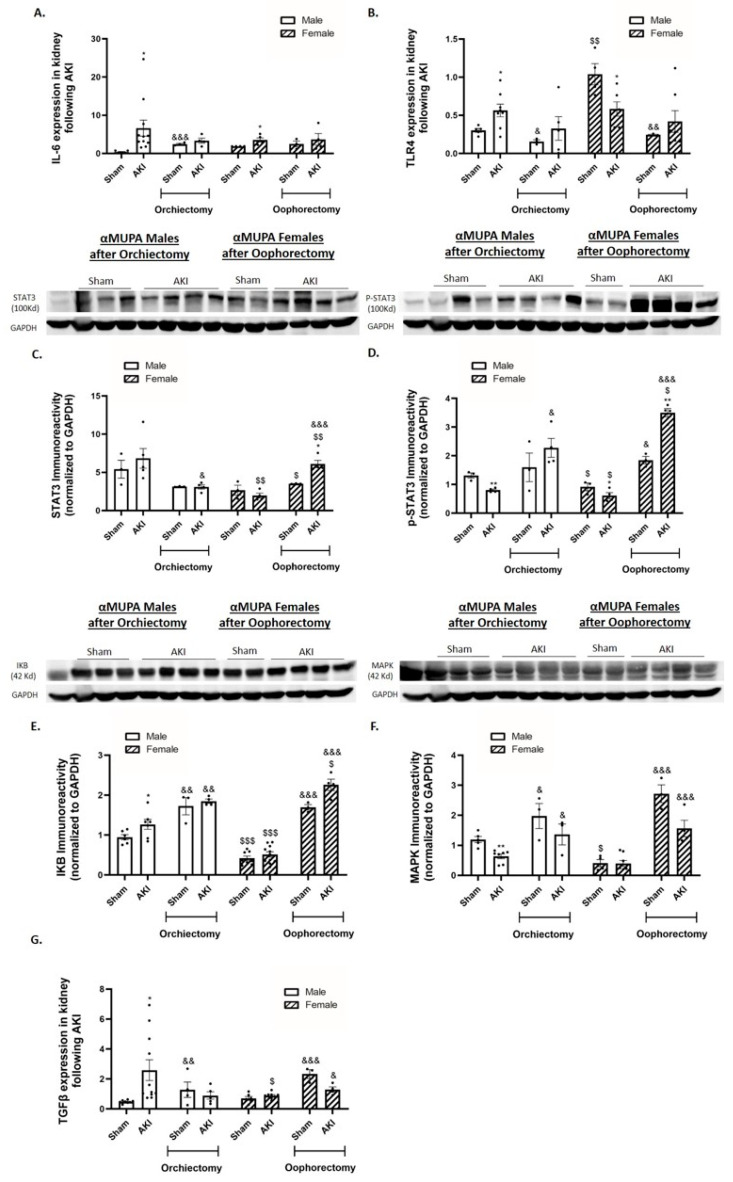
The impact of AKI on renal expression/abundance of inflammatory and fibrotic markers in αMUPA male and female mice before and after gonadectomy/L-NAME treatment. (**A**) Expression of IL-6; (**B**) expression of Toll like receptor 4 (TLR4); (**C**) immunoreactive levels of STAT3; (**D**) immunoreactive levels of p-STAT3; (**E**) immunoreactive levels of IKB; (**F**) immunoreactive levels of MAPK; and (**G**) expression of TGFβ following orchiectomy (males) or oophorectomy (females). (**H**) Expression of IL-6; (**I**) expression of Toll like receptor 4 (TLR4); (**J**) immunoreactive levels of STAT3; (**K**) immunoreactive levels of p-STAT3; (**L**) immunoreactive levels of IKB; (**M**) immunoreactive levels of MAPK; and (**N**) expression of TGFβ after L-NAME treatment. (* *p* < 0.05, ** *p* < 0.01, —sham vs. AKI in the same group, ^$^
*p* < 0.05, ^$$^
*p* < 0.01, ^$$$^ *p* < 0.001—male vs. female, which underwent similar procedure/treatment, ^&^ *p* < 0.05, ^&&^ *p* < 0.01, ^&&&^ *p* < 0.001—before vs. after orchiectomy (males) or oophorectomy (females)/L-NAME treatment).

**Figure 6 ijms-25-03544-f006:**
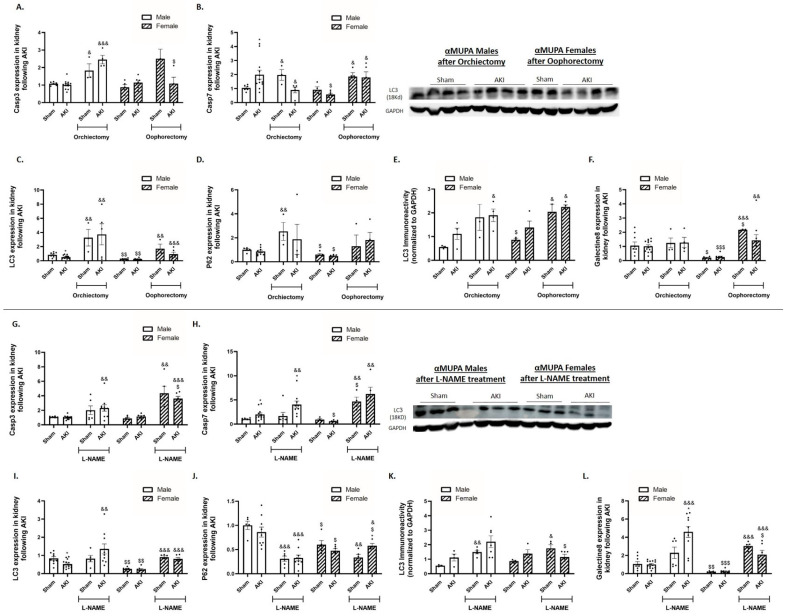
Impact of AKI on renal expression/abundance of apoptotic and autophagy markers in αMUPA male and female mice before and after gonadectomy/L-NAME treatment. (**A**) Expression of Caspase 3; (**B**) expression of Caspase 7; (**C**) expression of LC3; (**D**) expression of P62; (**E**) immunoreactive levels of LC3; and (**F**) expression of Galectin 8 following orchiectomy (males) or oophorectomy (females). (**G**) Expression of Caspase 3; (**H**) expression of Caspase 7; (**I**) expression of LC3; (**J**) expression of P62; (**K**) immunoreactive levels of LC3; and (**L**) expression of Galectin 8 after L-NAME treatment. (* *p* < 0.05—sham vs. AKI in the same group, ^$^ *p* < 0.05, ^$$^ *p* < 0.01, ^$$$^ *p* < 0.001—male vs. female, which underwent similar procedure/treatment, ^&^ *p* < 0.05, ^&&^ *p* < 0.01, ^&&&^ *p* < 0.001—before vs. after orchiectomy (males) or oophorectomy (females)/L-NAME treatment).

**Figure 7 ijms-25-03544-f007:**
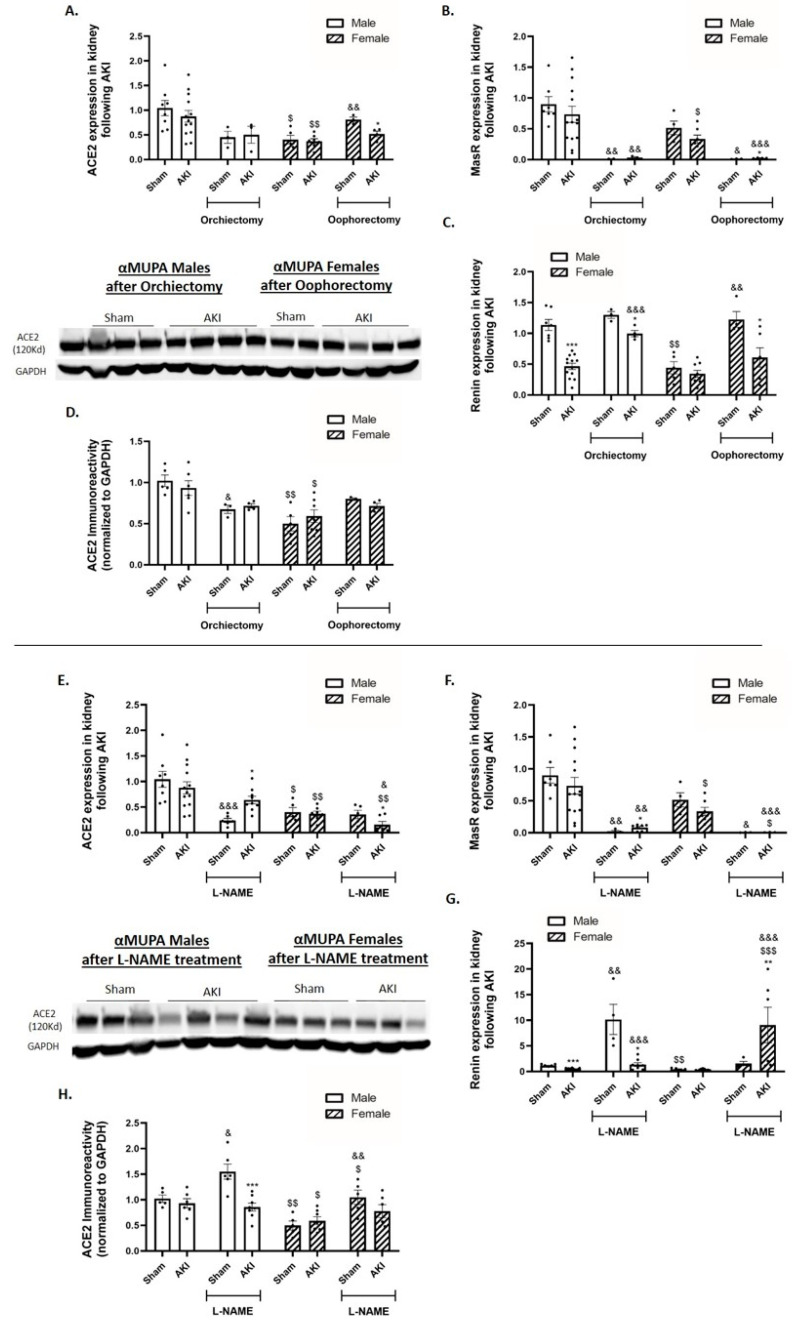
Effect of AKI on renal ACE2, MasR and renin in αMUPA male and female mice before and after gonadectomy/L-NAME treatment. (**A**) Expression of ACE2; (**B**) expression of Mas receptor; (**C**) expression of renin; and (**D**) ACE2 immunoreactive levels amount following orchiectomy (males) or oophorectomy (females). (**E**) Expression of ACE2; (**F**) expression of Mas receptor; (**G**) expression of renin; and (**H**) ACE2 immunoreactive levels amount after L-NAME treatment. (* *p* < 0.05, ** *p* < 0.01, *** *p* < 0.001—sham vs. AKI in the same group, ^$^ *p* < 0.05, ^$$^
*p* < 0.01, ^$$$^ *p* < 0.001—male vs. female, which underwent similar procedure/treatment, ^&^ *p* < 0.05, ^&&^ *p* < 0.01, ^&&&^ *p* < 0.001—before vs. after orchiectomy (males) or oophorectomy (females)/L-NAME treatment).

**Figure 8 ijms-25-03544-f008:**
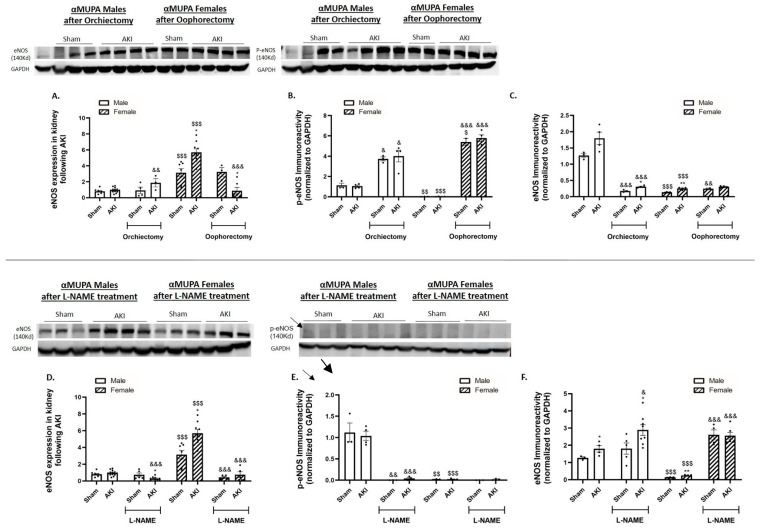
Effect of AKI on renal expression of eNOS and p-eNOS in αMUPA male and female mice before and after gonadectomy/L-NAME treatment. (**A**) eNOS immunoreactivity; (**B**) p-eNOS immunoreactivity; and (**C**) expression of eNOS following orchiectomy (males) or oophorectomy (females). (**D**) eNOS immunoreactivity; (**E**) p-eNOS immunoreactivity; and (**F**) expression of eNOS after L-NAME treatment. (* *p* < 0.05, ** *p* < 0.01—sham vs. AKI in the same group, ^$$^ *p* < 0.01, ^$$$^ *p* < 0.001—male vs. female, which underwent similar procedure/treatment, ^&^ *p* < 0.05, ^&&^ *p* < 0.01, ^&&&^ *p* < 0.001—before vs. after orchiectomy (males) or oophorectomy (females)/L-NAME treatment).

**Table 1 ijms-25-03544-t001:** Effects of AKI on body and kidney weight in αMUPA mice following Gonadectomy/L-NAME treatment.

Group	Body Weight before Surgery (g.)	Body Weight after Surgery (g.)	% Change in BW	Kidney Weight (KW) (g.)	Kidney/Body Weight Ratio (%)
αMUPA male—Sham	23.76 ± 0.57	23.94 ± 0.8	+0.75%	0.16 ± 0.005	0.68 ± 0.01
αMUPA male—AKI	23.97 ± 0.41	21.97 ± 0.3	−8.34% ^###^	0.174 ± 0.005	0.79 ± 0.02 **
αMUPA male—Sham after orchiectomy	25.48 ± 0.45	24.4 ± 0.77	−4.2%	0.13 ± 0.003	0.54 ± 0.02 ^&&&^
αMUPA male—AKI after orchiectomy	25.17 ± 0.42	23.68 ± 0.39	−5.9%	0.142 ± 0.003	0.6 ± 0.003 *,^&&&^
αMUPA male—Sham after L-NAME treatment	27.11 ± 0.4	26.92 ± 0.34	−0.7%	0.174 ± 0.005	0.64 ± 0.01 ^&^
αMUPA male—AKI after L-NAME treatment	25.97 ± 0.53	24.16 ± 0.41	−6.97% ^#^	0.185 ± 0.005	0.76 ± 0.01 ***,^&^
αMUPA female—Sham	19.76 ± 0.83	19.85 ± 1.1	+0.45%	0.11 ± 0.005	0.53 ± 0.02 ^$$$^
αMUPA female—AKI	20.71 ± 0.48	19.66 ± 0.49	−5%	0.117 ± 0.005	0.62 ± 0.02 *,^$$$^
αMUPA female—Sham after oophorectomy	21.81 ± 1.12	21.2 ± 0.94	−2.8%	0.124 ± 0.012	0.58 ± 0.02
αMUPA female—AKI after oophorectomy	21.7 ± 0.49	19.92 ± 0.5	−8.2% ^#^	0.132 ± 0.005	0.68± 0.02 *,^&^,^$$^
αMUPA female—Sham after L-NAME treatment	22.2 ± 0.8	21.36 ± 0.7	−3.7%	0.123 ± 0.002	0.57 ± 0.007 ^$^
αMUPA female—AKI after L-NAME treatment	22 ± 0.52	20.25 ± 0.49	−7.9% ^#^	0.139 ± 0.006	0.68 ± 0.01 **,^$^,^&^

Body and kidney weight of αMUPA male and female mice, before and after AKI, following orchiectomy (males) or oophorectomy (females) or L-NAME treatment. ^#^ *p* < 0.05, ^###^ *p* < 0.001—body weight before vs. body weight after AKI, * *p* < 0.05, ** *p* < 0.01, *** *p* < 0.001—Sham vs. AKI in the same group, ^$^ *p* < 0.05, ^$$^ *p* < 0.01, ^$$$^ *p* < 0.001—male vs. female, which underwent similar procedure/L-NAME treatment, ^&^ *p* < 0.05, ^&&&^ *p* < 0.001—before vs. after orchiectomy (males) or oophorectomy (females)/L-NAME treatment.

## Data Availability

The data that support the findings of this study are available on request from the corresponding author (ZA).

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
