# Peer review of "Gender-Specific Renoprotective Pathways in αMUPA Transgenic Mice Subjected to Acute Kidney Injury"

_ijms, 2024, doi:10.3390/ijms25063544_

Round 1

Reviewer 1 Report

Comments and Suggestions for Authors

The authors have conducted an insightful study on female nephroprotection, a topic of great value and interest in the field of nephrology. The manuscript is coherent, and at the experimental level, they have carried out a sufficient number of experiments to support the working hypothesis. However, in order to enhance the quality of the article, I suggest the following modifications or recommendations:

Statistical Tests: The authors should justify the use of parametric statistics by performing the corresponding normality tests and providing a rationale for an appropriate sample size. Furthermore, when making comparisons between various experimental groups, statistical tests such as one-way ANOVA or Kruskal-Wallis should be conducted. In some comparative contexts, even two-way ANOVA might be necessary.

Graphical Representation of Results: Following statistical recommendations, consider making changes to the graphical representation of results. Specify in the figure caption how the data have been represented (mean, median, etc.), and contemplate including a representation of individual values (using dots for each value) to enhance result transparency and overall scientific quality.

Western Blot: Consider including the molecular weight and a reference to the molecular weights of the marker used in the technique. Additionally, it is customary to include complete membranes in the supplementary material section to ensure the proper execution of the technique and to demonstrate whether nonspecific bands are also produced.

Author Response

We thank reviewer 1 for his/her valuable comments.

Please find bellow itemized response to these comments:

The authors have conducted an insightful study on female nephroprotection, a topic of great value and interest in the field of nephrology. The manuscript is coherent, and at the experimental level, they have carried out a sufficient number of experiments to support the working hypothesis. However, in order to enhance the quality of the article, I suggest the following modifications or recommendations:

Comment 1: Statistical Tests: The authors should justify the use of parametric statistics by performing the corresponding normality tests and providing a rationale for an appropriate sample size. Furthermore, when making comparisons between various experimental groups, statistical tests such as one-way ANOVA or Kruskal-Wallis should be conducted. In some comparative contexts, even two-way ANOVA might be necessary.

Response: Thank you for the note. The sample size was determined based on statistical power. Specifically, n in each group was determined to achieve 80% power with α=0.05 (see p.7). Concerning the statistical tests that were applied, statistical significance was tested for comparisons between male and female αMUPA mice that underwent either sham or AKI by using unpaired Student’s t-tests. In addition, we compared between males and females before and after gonadectomy. Since we compared only between two groups at the same time, t-test is suitable for such comparison. We consulted professional statistician and he confirmed the currently applied tests.  

Comment 2: Graphical Representation of Results: Following statistical recommendations, consider making changes to the graphical representation of results. Specify in the figure caption how the data have been represented (mean, median, etc.), and contemplate including a representation of individual values (using dots for each value) to enhance result transparency and overall scientific quality.

Response: Thank you for the note. We improve the quality of the figures. We understand the difficulties in following up the applied comparisons between the various groups. However, since we have 8 groups, applying scattered data will create crowded figures which already overwhelming. It should be emphasized that the SEM was low due to minor variability between the repeated samples of the same group. We use * to compare between sham and AKI regardless the gender and the strain, $ to compare between males and females in the same strain that underwent the same procedure, and & to compare between MUPA mice of the same gender with and without gonadectomy. We applied this clue in all the relevant legends.

Comment 3: Western Blot: Consider including the molecular weight and a reference to the molecular weights of the marker used in the technique. Additionally, it is customary to include complete membranes in the supplementary material section to ensure the proper execution of the technique and to demonstrate whether nonspecific bands are also produced.

Response: We did our best to improve the quality of the western blots and added the molecular weight of the relevant bands (see fig. 3-8). We would like to emphasize that we examined many samples for each studied parameter, therefore the graphs are crowded, yet we combined representative gels along quantification bars. We uploaded the original gels on the submission website.

Reviewer 2 Report

Comments and Suggestions for Authors

Thank you for submitting the manuscript " Gender-Specific Renoprotection Pathways in αMUPA Trans-2 genic Mice Subjected to Acute Kidney Injury" to IJMS. The research reported in the manuscript is interesting and important from a scientific point of view to address the gender specific differences in response to AKI and understand the differential regulation mechanistic pathways. Based on authors previous findings that female αMUPA mice protected against AKI as compared to males as results clearly indicated and supporting their hypothesis that sex hormones could potentially play a role in this phenomenon, where estrogen exerts a beneficial and testosterone adverse role in I/R-induced AKI model. The manuscript is well described including the methods and discussion.

I have very minor comment.

Authors have to indicate histology quantification in Figure 2.

Author Response

We thank reviewer 2 for his/her valuable comments.

Please find bellow itemized response to these comments:

Thank you for submitting the manuscript " Gender-Specific Renoprotection Pathways in αMUPA Trans-2 genic Mice Subjected to Acute Kidney Injury" to IJMS. The research reported in the manuscript is interesting and important from a scientific point of view to address the gender specific differences in response to AKI and understand the differential regulation mechanistic pathways. Based on authors previous findings that female αMUPA mice protected against AKI as compared to males as results clearly indicated and supporting their hypothesis that sex hormones could potentially play a role in this phenomenon, where estrogen exerts a beneficial and testosterone adverse role in I/R-induced AKI model. The manuscript is well described including the methods and discussion.

I have very minor comment.

Comment 1: Authors have to indicate histology quantification in Figure 2.

Reply: Thank you for the note. Indeed, quantification of the histological changes will give more accurate magnitude of the renal injury, as compared with the visual impression as we indicted in our MS. However, the differences between sham and AKI and between males and females as well as the impact of gonadectomy or L-NAME treatment are so obvious.     

Reviewer 3 Report

Comments and Suggestions for Authors

The article “Gender-Specific Renoprotection Pathways in αMUPA Transgenic Mice Subjected to Acute Kidney Injury” presents the impact of sex differences on renal protection in the context of acute kidney injury (AKI) in αMUPA transgenic mice. The study examined the effects of gonadectomy (orchiectomy and oophorectomy) and L-NAME treatment on the renal response to AKI.

The paper is well structured with clear sections including introduction, methods, results and discussion. However, some sections, especially in the results, are dense and would benefit from additional subheadings or bullet points for easier understanding.

In terms of methodology, the experimental design appears to be sound and includes a detailed description of the procedures and controls used. Nevertheless, a more detailed explanation of the choice of the αMUPA transgenic mouse model and its significance for human physiology would enhance the reader's understanding of the implications of the study.

The presentation of the study results itself also seems detailed, however, in my opinion the article could be improved by adding more graphical representations of the data, such as graphs, to help visualize the observed differences and trends. Additionally, it seems that some of the figures in the article are not of high quality, which makes it difficult to interpret the results they present. Improving the resolution and clarity of this data would greatly improve the reader's ability to understand the data. Moreover, when it comes to statistics, in my opinion, a more detailed explanation of statistical tests and justification of their choice would be useful.

The discussion presented in the manuscript quite comprehensively covers the implications of the findings and compares them to the existing literature. However, again, despite being prepared in this way, in my opinion the quality of the manuscript would have benefited from a more explicit statement regarding the limitations of the study and potential areas for future research. At the same time, I ask the authors to consider whether their conclusions are not too speculative.

To sum up, the article is prepared from a technical point of view, its overall readability is good and it should reach an industry audience. However, in the above context, I have one final comment. In my opinion, and in line with the trend of open science, this article should be lightly edited to make it more accessible to a wider audience by simplifying complex jargon where possible, without compromising scientific integrity.

Finally, after taking into account my small corrections and suggestions, I am definitely in favor of publishing this work.

Author Response

We thank reviewer 3 for his/her valuable comments.

Please find bellow itemized response to these comments:

The article “Gender-Specific Renoprotection Pathways in αMUPA Transgenic Mice Subjected to Acute Kidney Injury” presents the impact of sex differences on renal protection in the context of acute kidney injury (AKI) in αMUPA transgenic mice. The study examined the effects of gonadectomy (orchiectomy and oophorectomy) and L-NAME treatment on the renal response to AKI.

The paper is well structured with clear sections including introduction, methods, results and discussion. However, some sections, especially in the results, are dense and would benefit from additional subheadings or bullet points for easier understanding.

Comment 1: In terms of methodology, the experimental design appears to be sound and includes a detailed description of the procedures and controls used. Nevertheless, a more detailed explanation of the choice of the αMUPA transgenic mouse model and its significance for human physiology would enhance the reader's understanding of the implications of the study.

Response: Thank you for calling our attention to this issue. αMUPA mice are characterized  by turning uPA over-expression in the trigeminal nucleus in the brain. These mice are resistant to obesity, which is known as a risk factor for cardiovascular and renal diseases. Moreover, experimental studied showed that αMUPA mice were protected against myocardial infarction (MI) [1,2], sepsis [3] as well as after AKI induction [4]. In addition, urokinase plasminogen activator (uPA) is a widely used drug for treatment of clots. In addition, several studies showed that it is involved in the breakdown of the extracellular matrix and in angiogenesis [5].

Upon your request, we added description of MUPA mice and the sense behind applying this model of mice in the current study (see p.6).

References:

  1. Levy E, Kornowski R, Gavrieli R, Fratty I, Greenberg G, Waldman M, Birk E, Shainberg A, Akirov A, Miskin R, Hochhauser E. Long-Lived alphaMUPA Mice Show Attenuation of Cardiac Aging and Leptin-Dependent Cardioprotection. PLoS One. 2015; 10: e0144593.
  2. Abd Alkhaleq H, Kornowski R, Waldman M, Levy E, Zemel R, Nudelman V, Shainberg A, Miskin R, Hochhauser E. Leptin modulates gene expression in the heart and cardiomyocytes towards mitigating ischemia-induced damage. Exp Cell Res. 2020; 397: 112373.
  3. Abd Alkhaleq H, Kornowski R, Waldman M, Zemel R, Lev DL, Shainberg A, Miskin R, Hochhauser E. Leptin modulates gene expression in the heart, cardiomyocytes and the adipose tissue thus mitigating LPS-induced damage. Exp Cell Res. 2021; 404: 112647.
  4. Alkhaleq HA, Karram T, Fokra A, Hamoud S, Kabala A, Abassi Z. The Protective Pathways Activated in Kidneys of alphaMUPA Transgenic Mice Following Ischemia\Reperfusion-Induced Acute Kidney Injury. Cells. 2023; 12.
  5. Berkenblit A, Matulonis UA, Kroener JF, Dezube BJ, Lam GN, Cuasay LC, Brunner N, Jones TR, Silverman MH, Gold MA. A6, a urokinase plasminogen activator (uPA)-derived peptide in patients with advanced gynecologic cancer: a phase I trial. Gynecol Oncol. 2005; 99: 50-7.

Comment 2: The presentation of the study results itself also seems detailed, however, in my opinion the article could be improved by adding more graphical representations of the data, such as graphs, to help visualize the observed differences and trends. Additionally, it seems that some of the figures in the article are not of high quality, which makes it difficult to interpret the results they present. Improving the resolution and clarity of this data would greatly improve the reader's ability to understand the data. Moreover, when it comes to statistics, in my opinion, a more detailed explanation of statistical tests and justification of their choice would be useful.

Response: Thank you for the note. We improve the quality of the figures included the western blot representatives and the relevant quantification figures. Moreover, figure 5, which was remarkably crowed, was split into two subfigures (Figure 5A and 5B).

We understand the difficulties in following up the applied comparisons between the various groups and the differences and trends. However, since we have eight groups and eight figures, adopting additional presentation data will create overwhelming number of figures, which already are crowded.

We used * to compare between sham and AKI regardless the gender and the strain, $ to compare between males and females in the same strain that underwent the same procedure, and & to compare between MUPA mice of the same gender with and without gonadectomy. We added this clue in all the relevant figure legends. In addition, the various comparisons that were applied now are described under “statistical analysis” session (see p. 11).

Comment 3: The discussion presented in the manuscript quite comprehensively covers the implications of the findings and compares them to the existing literature. However, again, despite being prepared in this way, in my opinion the quality of the manuscript would have benefited from a more explicit statement regarding the limitations of the study and potential areas for future research. At the same time, I ask the authors to consider whether their conclusions are not too speculative.

Response: Thank you for the note. We added at the end of the discussion the limitation of the current study (see p. 23) and soften our conclusions (see abstract).

To sum up, the article is prepared from a technical point of view, its overall readability is good and it should reach an industry audience. However, in the above context, I have one final comment. In my opinion, and in line with the trend of open science, this article should be lightly edited to make it more accessible to a wider audience by simplifying complex jargon where possible, without compromising scientific integrity.

Finally, after taking into account my small corrections and suggestions, I am definitely in favor of publishing this work.

Response: Thank you.

Round 2

Reviewer 3 Report

Comments and Suggestions for Authors

The authors made corrections and took into account suggestions. For my part, the current version of the article is acceptable for publication.

Author Response

(The authors gave the same response as above.)
